# Jet: A Modern Transformer-Based Normalizing Flow

**Alexander Kolesnikov**[*]                                                    *akolesnikov@google.com*
*Google DeepMind*

**André Susano Pinto**[*]                                                      *andresp@google.com*
*Google DeepMind*

**Michael Tschannen**[*]                                                       *tschannen@google.com*
*Google DeepMind*

**Reviewed on OpenReview:** *https: // openreview. net/ forum? id= jdvnaki7ZY*

## Abstract

In the past, normalizing generative flows have emerged as a promising class of generative models for natural images. This type of model has many modeling advantages: the ability to efficiently compute log-likelihood of the input data, fast generation, and simple overall structure. Normalizing flows remained a topic of active research but later fell out of favor, as visual quality of the samples was not competitive with other model classes, such as GANs, VQ-VAE-based approaches or diffusion models. In this paper we revisit the design of coupling-based normalizing flow models by carefully ablating prior design choices and using computational blocks based on the Vision Transformer architecture, not convolutional neural networks. As a result, we achieve a much simpler architecture that matches existing normalizing flow models and improves over them when paired with pretraining. While the overall visual quality is still behind the current state-of-the-art models, we argue that strong normalizing flow models can help advancing the research frontier by serving as building components of more powerful generative models.

## 1 Introduction

In this paper we explicitly do not attempt to devise the new state-of-the art image modeling approach or propose a new paradigm. Instead, we revisit the long known but recently neglected class of models for generative modeling: coupling-based normalizing flows. Normalizing flows have important capabilities that make them a useful tool for modern generative modeling.

At a high-level, a normalizing flow model learns a bijective (and thus invertible) mapping $g$ from the input space to the latent space, where the latent space follows a simple distribution, e.g. Gaussian distribution. A complex bijective transformation $g$ can be constructed by stacking multiple *coupling blocks*, which are bijective and invertible in closed form by design and are parametrized by deep neural networks.

Normalizing flow models can be directly trained by computing data log-likelihood in the simple (e.g. Gaussian) latent space after the learnable and differentiable mapping $g$ is applied to transform training examples. For data generation, the inverse transformation $g^{-1}$ is readily available, which can be used to map easy-to-sample Gaussian latent space to the samples from the target distribution. The two explicit, differentiable and lossless mappings $g$ and $g^{-1}$ can be used as building blocks for more complex generative systems. For example, normalizing flows are used as a critical component of more complex systems: recent examples include (Chen et al., 2016; Kingma et al., 2016; Tschannen et al., 2024a) and (Tschannen et al., 2024b) that leverage normalizing flows to facilitate image modeling with VAEs and autoregressive transformers respectively. This motivates us to revisit the normalizing flow model class.

---

[*]Equal contribution.

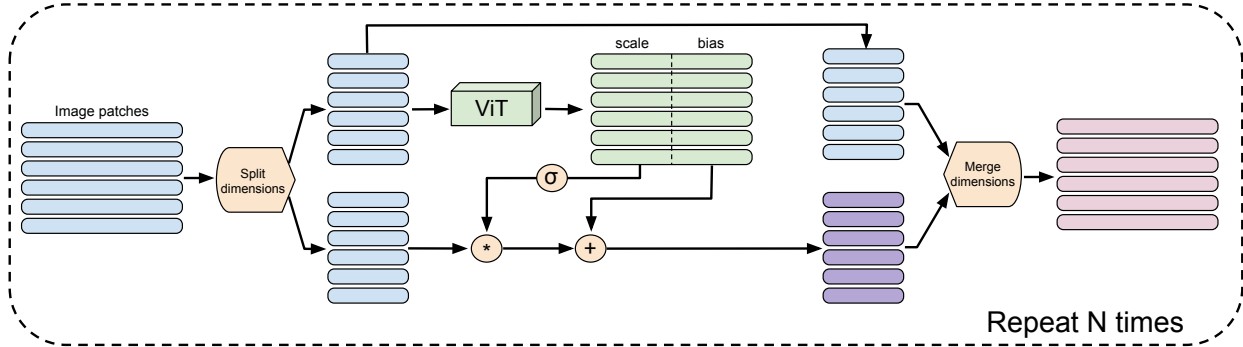

Figure 1: Overview of the Jet model. The dashed box contains a coupling layer computing an affine transform from one half of the input dimensions (patches or features) and then applying it to the other half of the input dimensions. The full model is obtained by stacking $N$ such invertible coupling layers.

In the mid-to-late 2010s normalizing flow models were a topic of active research. NICE (Dinh et al., 2014) was the early normalizing flow model for images. It introduced the main building block behind normalizing flow models: an additive coupling layer. RealNVP (Dinh et al., 2017) improved over NICE by introducing multiscale architecture and affine coupling layers that additionally perform a scaling transformation. Subsequently, in Glow (Kingma & Dhariwal, 2018), authors introduce two more components: an invertible dense layer and specialized activation normalization layer. Finally, Flow++ (Ho et al., 2019) shows improvements from using dequantization flow trick and generalized variant of the affine coupling block.

In this paper we mainly concentrate on revisiting optimal design for the normalizing flow models. We focus on both performance and simplicity of the final model. We built on top of the prior literature and put all components under a careful scrutiny. Our final model has a radically simpler design and only relies on the plain affine coupling blocks parametrized by the Vision Transformer model. Our key contributions can be summarized as follows:

- We use Vision Transformer building blocks instead of convolutional neural networks, which leads to a significant performance improvement.

- We drastically simplify the overall architecture by eliminating many components from prior models:
  - No multiscale components and early factored-out channels
  - No invertible dense layers
  - No "activation normalization" layers
  - No dequantization flow
  - No generalized coupling transformation

- We show the proposed architecture matches SOTA results in terms of negative log-likelihood on common image benchmarks when trained on the same data-limited datasets.

- We demonstrate that pretraining a Jet model on a large corpus of natural images is highly effective and simple, and that it surpasses prior state-of-the-art results.

## 2 Method

In the section we introduce the Jet model. We first introduce the architecture, then describe training procedure and important implementation details.

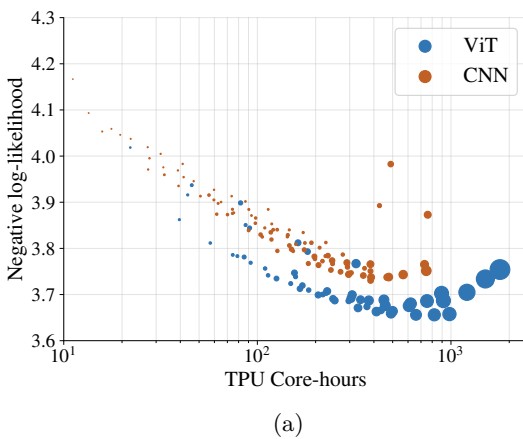 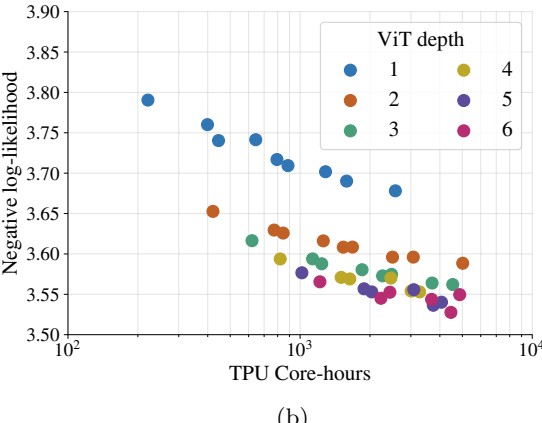

(a)  (b)

Figure 2: Effect of different architecture design choices on the validation NLL (in bits per dimension), as a function of training compute. Figure 2a: Results on ImageNet-1k $64 \times 64$ for CNN vs ViT blocks (the marker size is proportional to the model parameter count). ViT blocks clearly outperform CNN blocks for a given training compute budget. Figure 2b: Results on ImageNet-21k $32 \times 32$ for different ViT depths. Increasing the block depth leads to improved results up to depth 5.

## 2.1 Jet model

The Jet model has a very simple high-level structure. First, the input image is split into $K$ flat patches, flattened into a sequence of vectors, and then we repeatedly apply affine coupling layers (Dinh et al., 2017). We illustrate the Jet model architecture in Figure 1.

The input to a coupling layer is $x \in \mathbb{R}^{K \times 2d}$, where $K$ is the total number of patches and $2d$ is the number of dimensions within each patch after flattening. We then apply a dimension splitting transformation, producing $x_1, x_2 \in \mathbb{R}^{K \times d}$. The first part, $x_1$, is not modified, yielding $y_1$. The second part, $x_2$ is modified with element-wise affine transformation $y_2 = (x_2 + b) \cdot s$, where $s, b \in \mathbb{R}^{K \times d}$. Scaling factor $s$ and bias $b$ are functions of $x_1$. We use a single Vision Transformer (ViT) backbone, $f : \mathbb{R}^{K \times d} \to \mathbb{R}^{K \times 2d}$, which maps $x_1$ to scale and bias, effectively implementing the functions $s(x_1)$ and $b(x_1)$. Finally, we merge $y_1$ and $y_2$ by applying the inverse of the splitting transformation that was applied to $x$, and arrive at the output $y \in \mathbb{R}^{K \times 2d}$.

To stabilize training we additionally apply a sigmoid transformation, $\sigma$, that squashes the scale coefficient to $(0, 1)$ range and then multiply it by a fixed constant $m$ to extend the range to $(0, m)$. In practice we always set $m = 2$, which allows to amplify or suppress the intermediate activations, while keeping activations in a reasonable range when stacking many coupling layers.

Overall, a single coupling block can be formalized as follows:

$$y_1 = x_1$$
$$y_2 = (x_2 + b(x_1)) \cdot \sigma(s(x_1)) \cdot m$$

As we show below in Section 2.3, this transformation can be inverted for arbitrary functions $b(x_1)$ and $s(x_1)$.

## 2.2 Training and optimization objective.

We now recall the fundamentals of the normalizing flow model training. The key assumption behind the normalizing flows is that the target density $p(x)$ can be modeled as a simple distribution, $p_{\mathcal{N}}(z)$ (e.g. standard Gaussian) after using a bijective transformation $g : x \to z$ to transform the original data. Applying the "change of variable" identity from the basic probability calculus we obtain a tractable model for the

probability density function:

$$p(x) = p_{\mathcal{N}}(g(x)) \left| \det \frac{\partial g(x)}{\partial x^\top} \right|.$$

We implement function $g$ with the Jet model described above. The determinant of the derivative, $\frac{\partial g(x)}{\partial x^\top}$, is equal to the product of determinant of the derivatives of individual coupling layers. And for each coupling block the determinant computation is trivial and equal to the product of all scaling factors in the affine transform. See (Dinh et al., 2017) for details on the derivation. In practice we maximize data log-likelihood, so the optimization objective is

$$-\frac{1}{2} \sum_i \left[ g(x)_i^2 + \log(2\pi) \right] + \sum_\ell \sum_i \left[ \log \sigma(s_i^\ell) + \log m \right],$$

where we use index $i$ to iterate over all dimensions of an array inside the sum, and upper index $\ell$ to iterate over total number of blocks. The first sum term arises from taking the logarithm of the standard Gaussian density, applied to every output dimension. The second sum term is the log determinant, which is computed as sum of logarithms of all scale values across all layers, plus logarithm of the fixed $m$ multiplier.

In practice we normalize the above objective by the total number of dimensions and we keep all constant terms. Even though it does not affect optimization, by doing this we obtain a log-likelihood estimate, which can be interpreted as "bit-per-dimension" (bpd) (assuming a logarithm base of 2). For example, if the objective value is 3.1, it means that the average uncertainty per input dimension is 3.1 bits. A uniformly random image model should yield approximately 8 bpd. This is useful when comparing models across different modeling classes and for various correctness checks.

Note that to correctly model discrete densities (e.g. images where pixels are discrete-valued) we employ a standard dequantization procedure described in (Theis et al., 2015). In practice, it means that we add a random $[0, 1]$ uniform noise to input images, which consist of discrete values in the range $[0, 255]$. We perform this dequantization procedure both during training and evaluation.

### 2.3 Inverse transformation and image generation

It is easy to obtain the inverse of the above transformation in closed form. Notably, the ViT function that computes bias and scale terms does not need to be inverted:

$$\begin{aligned} x_1 &= y_1 \\ x_2 &= \frac{y_2}{\sigma(s(x_1)) \cdot m} - b(x_1) \end{aligned}$$

The computational complexity of computing the inverse is exactly the same as computing the normalizing flow itself. New images can be sampled by first sampling the target density (i.e. Gaussian noise) and then applying the inverse transformation.

### 2.4 Initialization

Careful initialization is essential for training a deep normalizing flow model with a large number of coupling blocks. We employ a simple yet very effective initialization scheme. The final linear projection of the ViT $f$ is initialized with zero weights. As a result, predicted bias values, $b(x_1)$ are 0. The scale values are equal to $\sigma(0) = 0.5$. When we set $m = 2$, then the all scaling factors become equal to 1.

As a result, the Jet model behaves as identity function at initialization. Empirically, we find this sufficient to ensure stable optimization at the beginning of training. Consequently, we do not need to add "ActNorm" layer that is commonly used in the normalizing flow literature to achieve a similar effect.

## 2.5 Dimension splitting

We explore various options for channel splitting. One option is to perform channel-wise split, by splitting the channels of each image patch into the two equal groups (Dinh et al., 2014; Kingma & Dhariwal, 2018). The splitting is random within each coupling layer and is fixed ahead of time (independently for each layer). This is a simple-to implement-strategy that ensures diverse channel mixing.

We also implement various splitting strategies to facilitate spatial mixing. To this end, we explore 3 strategies that respect images' 2D topology: row-wise alternating patch splitting, column-wise alternating patch-splitting, and the "checkerboard" splitting (Dinh et al., 2017). See Section 3 for the empirical investigation of performance for various design choices of splitting operations.

**Splitting implementation details.** The natural way to perform channel splitting is to use array indexing operations. However, indexing can be slower than matrix multiplication on modern accelerators. Thus, we implement channel splitting as matrix multiplication with precomputed 0 and 1 (frozen) weight matrices. A nice by-product of this approach is that dimension merging (inverse of splitting) can be trivially implemented as matrix multiplications with the transposed weight matrices.

**Numerical precision considerations.** Matrix multiplications on modern accelerators often implicitly uses half-precision for multiplying individual values, while accumulating the result using full `float32` precision. We note that such loss in precision may lead to numerical issues, as, for example, the uniform $[0, 1]$ dequantization noise loses most of its entropy and leads to overly optimistic log-likelihood estimates. To avoid this, we enable full precision matrix multiplication mode when splitting/merging the dimensions. Note that we still use fast default settings when computing the ViT function $f$ inside coupling layers. So the overall efficiency drop from enabling full precision for splitting/merging is negligible overall.

## 3 Experiments

Throughout the paper, we keep our experimental setup simple and unified. Additionally, the code is available in the `big_vision` codebase[1]. For the ViT architecture, we follow the original paper Dosovitskiy et al. (2021). Our ViT models inside the coupling layers do not have initial patchification or final pooling and linear projections. Unless stated otherwise, we set the patch size such that the total number of patches is equal to 256.

For the optimizer we use AdamW (Loshchilov et al., 2017). We set second momentum $\beta_2$ parameter to 0.95 to stabilize training. We use a cosine learning rate decay schedule.

**Datasets.** We perform experiments on three datasets: Imagenet-1k, Imagenet-21k and CIFAR-10, across two input resolutions: $32 \times 32$ and $64 \times 64$ (except for CIFAR-10). To downsample Imagenet-1k images we follow the standard protocol (Van den Oord et al., 2016b) to ensure a correct comparison to the prior art (i.e. we use the preprocessed data provided by (Van den Oord et al., 2016b) where available). To downsample Imagenet-21k images we use `TensorFlow` resize operation with method set to `AREA`[2]. For CIFAR-10 we use the original dataset resolution. Importantly, to make sure our results are comparable to the literature, we do not perform any data augmentations. For comparison with Denseflow (Grcić et al., 2021) which uses a different protocol see Appendix A.2.

### 3.1 Main results

We conduct extensive sweep that includes Jet models trained across varying computational capacity, data size (ImageNet-1k and ImageNet-21k) and resolutions ($32 \times 32$ and $64 \times 64$). For the model capacity sweep we explore the following configurations (approximately spanning 2 orders of magnitude in compute intensity):

- number of coupling layers in $\{16, 32, 64\}$ for ImageNet-1k and in $\{32, 64, 128\}$ for ImageNet-21k,
- the depth of ViT used inside the coupling layer in $\{1, 2, 3, 4, 5, 6\}$,

---

[1]https://github.com/google-research/big_vision
[2]https://www.tensorflow.org/api_docs/python/tf/image/resize

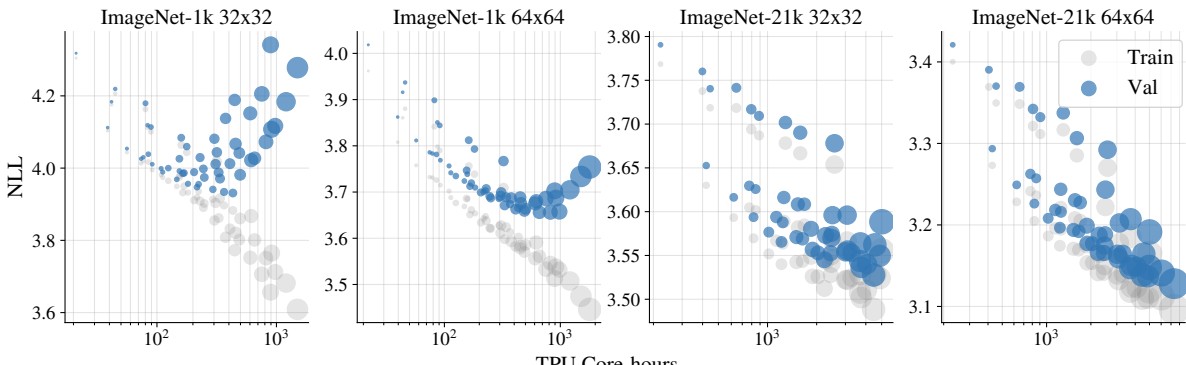

Figure 3: NLL as a function of training compute obtained when training Jet architectures with a range of architecture hyper-paramaters, for 4 different data sets. The size of each marker is proportional to the number of parameters in the model configuration. Overall we observe normalizing flow models benefit from scale, yet ImageNet-1k models start to overfit. When increasing the amount of data to ImageNet-21k size, we observe little overfitting and strong scaling trends.

Table 1: Negative log-likelihood of Jet models and the state-of-the-art coupling-based normalizing flow model Flow++ (Ho et al., 2019) from the literature. Jet (21k) indicates the result obtained when using a Jet model pretrained on ImageNet21k. Overall, we observe that all tasks benefit from this pretraining and that Jet matches or outperforms Flow++.

<div>

(a) ImageNet-1k $64 \times 64$

| Model | Result (NLL↓) |
|---|---|
| Flow++ | 3.69 |
| Jet | 3.656 |
| Jet (I21k) | 3.580 |

(b) ImageNet-1k $32 \times 32$

| Model | Result (NLL↓) |
|---|---|
| Flow++ | 3.86 |
| Jet | 3.931 |
| Jet (I21k) | 3.857 |

(c) CIFAR-10 $32 \times 32$

| Model | Result (NLL↓) |
|---|---|
| Flow++ | 3.08 |
| Jet (I21k) | 3.018 |

</div>

- the ViT embedding dimension in $\{256, 512, 768\}$ for ImageNet-1k and in $\{512, 768, 1024\}$ for ImageNet-21k. The number of heads is tied to the embedding dimension and equal to $\{4, 8, 12\}$ and $\{8, 12, 16\}$ respectively.

We use a fixed standard learning rate of 3e−4, weight decay of 1e−5 and train for 200 epochs for ImageNet-1k and for 50 epochs for ImageNet-21k. We additionally investigate transfer learning setup and finetune our best ImageNet-21k models on ImageNet-1k and CIFAR-10. Full sweep results are shown in Figure 3. Additionally, we present key numerical results in Table 1.

Our first observation is that due to the high expressive power of the Jet model parameterized by a ViT model, it tends to quickly overfit on ImageNet-1k, which has only 1.2M examples in total. However, despite this, for $64 \times 64$ Jet attains a state-of-the-art NLL of 3.66 bpd. For $32 \times 32$ input resolution overfitting for large models is more severe, however reasonably small models still achieve a competitive result of 3.93 bpd.

It is important to note that unlabeled natural images are abundant. Thus, the best way to tackle overfitting is to increase the amount of training data, as opposed to constraining or regularizing the model. We, therefore, use the Imagenet-21k dataset with more than 10× more images than in ImageNet-1k. As expected, overfitting is tamed, with larger models leading to increasingly better results.

The next important question is whether a model trained on a larger and more class-diverse ImageNet-21k dataset transfers to ImageNet-1k. We observe that with a very light finetuning (30 epochs, learning rate of 1e−5 and 3e−6 for higher resolution) we obtain state-of-the-art results on ImageNet-1k, attaining 3.58 and 3.86 bpd on $32 \times 32$ and $64 \times 64$ input resolution, respectively. However, one can argue that ImageNet-1k is very similar to ImageNet-21k and results are overly optimistic. To address this concern, we also transfer the

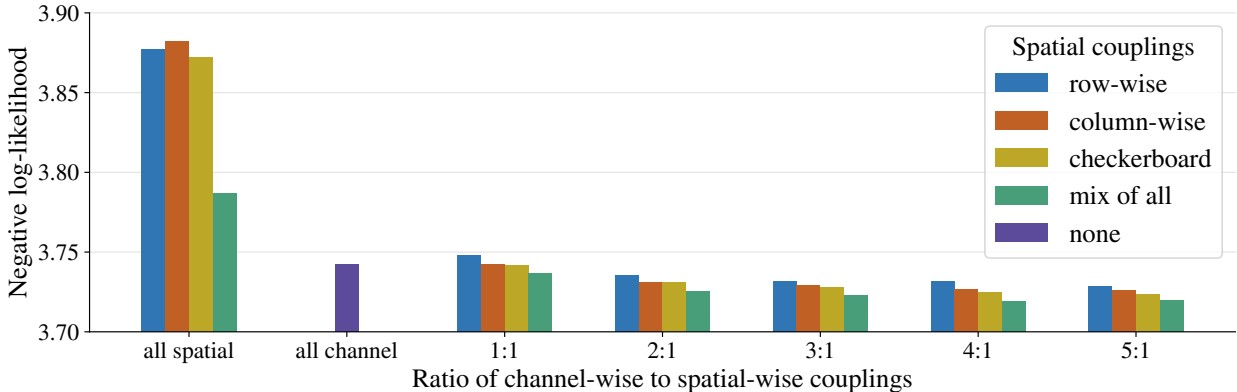

Figure 4: Ablation of coupling types. Negative log-likelihood on ImageNet-1k 64x64 when varying the ratio of channel-wise to spatial-wise couplings and when using different types of spatial-wise couplings. Results in table format in Appendix Table 10.

ImageNet-21k model to the CIFAR-10 dataset, which has very different type of images and has a different downsampling procedure. Again, after finetuning (100 epochs, learning rate of 3e−6) we obtain state-of-the-art performance of 3.02 bpd.

Overall, our results provide strong evidence that the Jet architecture achieves state-of-the art performance in the class of coupling-based normalizing flow models, and, at the same time, it is very stable and easy to train across wide range of scenarios.

## 3.2 Ablations

In this section we ablate key design choices for the Jet model. Our default ablation setting is moderately-sized Jet model, trained on ImageNet-1k $64 \times 64$ for 200 epochs. We set the total depth to 32 coupling layers, the ViT depth to 2 blocks and width to 512 dimensions. In our scaling study this configuration was reasonably close to the optimal setup, while being sufficiently fast for the extensive ablations sweeps. We use negative log-likelihood reported in bpd as the main ablation metric.

### 3.2.1 Coupling types

As presented in Section 2.5, we consider 4 different ways to split the channels in a coupling block: one is a channel-wise method (splits the channels into two parts) and three patch-wise methods (alternating rows, alternating columns or "checkerboard" splitting). Inspired by the prior literature, where channel-wise couplings were more prominent, we explore the following design space: $M$ repeated channel-wise couplings followed by 1 spatial coupling. The spatial coupling is either fixed to one of the methods, or alternates between 3 types. We vary $M$ from 0 to 5. Overall, the above choice space gives rise to 25 configurations.

Figure 4 presents the quantitative results, suggesting the following insights:

- Using interleaved spatial and channel coupling is optimal, where $M = 4$ is optimal, though $M \in 2..5$ achieves very similar performance.

- Alternating spatial coupling methods are superior over using any single coupling types.

- Spatial coupling alone is the worst configuration, while exclusive channel coupling is competitive.

### 3.2.2 Coupling layers vs ViT depth

Jet has two depth parameters: number of coupling layers and ViT depth within each coupling layer. The total compute is roughly proportional to the product of these two depths. We observe a very interesting

Table 2: Ablation of design choices reporting negative log-likelihood (bpd) on ImageNet-1k $64 \times 64$. Table 2a: shows the effect of using masking-mode or pairing-mode (best)when splitting the spatial tokens. Table 2b: shows the effect of using invertible dense layers and/or activation norm. Best results obtained when not using either.

(a)

| Couplings ratio | Masking | Pairing |
|---|---|---|
| all spatial | 3.844 | 3.787 |
| 1:1 | 3.741 | 3.737 |
| 2:1 | 3.727 | 3.725 |
| 3:1 | 3.723 | 3.723 |
| 4:1 | 3.722 | 3.719 |
| 5:1 | 3.722 | 3.720 |

(b)

| Activation Norm | Invertible Dense | NLL↓ |
|---|---|---|
| × | × | 3.720 |
| ✓ | × | 3.727 |
| × | ✓ | 3.741 |
| ✓ | ✓ | 3.733 |

interplay between these parameters, see Figure 2b representing ImageNet-21k $32 \times 32$, which is a detailed view of Figure 3.

Specifically, we observe that scaling the number of coupling layers, while keeping shallow ViT models (e.g. depth 1) results in an unfavorable compute-performance trade-off. It appears that ViT depth of at least 4–6 is the necessary condition for the Jet model to stay close to the frontier. For example, a model with 32 coupling layers and ViT depth 4 has roughly the same compute requirements as a model with 128 coupling layers and ViT depth 1. However, the former performs much better than the latter for fixed compute.

### 3.2.3 ViT vs CNNs

To ablate the use a ViT instead of a CNN block, we conduct a similar sweep to our main sweep on ImageNet-1k $64 \times 64$ but using a CNN architecture (specifically we use the CNN architecture from (Kolesnikov et al., 2020)). This time sweeping the following settings for the CNN setup: model depth in $\{16, 32, 64\}$, CNN block depth in $\{1, \ldots, 8\}$, block embedding dimension in $\{256, 512, 768, 1024, 1536\}$. Block dimension 1536 was not used for model depth 64 due to significant memory costs. The results in Figure 2a show that the CNN-based variant lags significantly behind the ViT-based one.

We anticipate that this gap can be reduced by using multiscale architectures, as commonly done in the literature (Dinh et al., 2017; Kingma & Dhariwal, 2018). However, in this paper we strive to simplify design and exclude multiscale architectures.

### 3.2.4 Coupling implementation

We investigate two common approaches for implementing coupling layers: masking or pairing. To be concrete, let's assume that we implement a coupling layer that splits the input spatially into two groups of patches. In one approach, which we name "masking" mode, we feed the $K$ patches to the ViT block but mask with zeros the ones corresponding to the $x_2$ group. At the output we use only the ones corresponding to the $x_2$ group and ignore the output of the patches corresponding to the $x_1$ group. One potential issue with this method is that it weakens the residual connections as the tokens from which we predict the output are zero tokens.

Another approach we consider is a "pairing" mode in which we establish a pairing between input and output patches (or embeddings). For example when using a vertical-stripes pattern, the outputs of the ViT block for a patch in the $x_1$ group will predict the scale and bias for a patch in the $x_2$ group (e.g. to the patch below). This would make the ViT block processes only $K/2$ patches.

We experiment with these two implementation types while sweeping the $M$:1 channel:spatial coupling ratios as in Section 3.2.1. The results presented in Table 2a indicate pairing to be superior, though the impact becomes smaller as one increases the number of channel couplings which do not depend on this design decision. We observe that both methods perform very similarly, with the pairing being slightly ahead of

masking, especially for the scenario when only spatial couplings are used. Thus, for the Jet model, we default to using pairing mode.

### 3.2.5   Invertible dense layers and activation normalization

Glow (Kingma & Dhariwal, 2018) introduces two components to improve the performance of normalizing flows: (1) a learnable, invertible dense layer which replaces the fixed permutation used to split the channels for each coupling; (2) an activation normalization layer with a scalar and bias parameters per channel similar to batch normalization.

We use neither of them in Jet, but ablate whether performance could be improved by introducing those in Table 2b. As a note, we observed that the use of activation normalization alone to be highly unstable, we sweep additional learning rate and seeds and report the best result found. Overall we found that not using any of those components leads to the best results.

### 3.2.6   Uniform dequantization vs dequantization flow

Flow++ (Ho et al., 2019) introduces a variational dequantization scheme to normalizing flows. Concretely, it proposes to replace the uniform dequantization noise added to the input with an image-conditional, learned noise distribution modeled by another normalizing flow. We ablate this component by training a 64 layer Jet model with 16-layer dequantization flow and compare it with an 80-layer base Jet model. The image conditioning of the dequantization flow was implemented by adding cross-attention layers to the ViT-blocks to the input. We observe no significant improvements when using the dequantization flow component.

## 4   Related work

NICE (Dinh et al., 2014) popularized coupling-based normalizing flows with the introduction of the additive coupling layer. RealNVP (Dinh et al., 2017) then increased the flow's expressivity by using affine coupling layers in combination with a multiscale architecture, and (Kingma & Dhariwal, 2018; Hoogeboom et al., 2019; Sukthanker et al., 2022) proposed additional specialized invertible layers for image modeling. Flow++ (Ho et al., 2019) demonstrated improvements from learning the dequantization noise distribution along with the flow model.

Another class of likelihood-based generative models are autoregressive models which flatten the (sub)pixels of an image into a sequence. Autoregressive modeling is enabled by using CNNs (Van den Oord et al., 2016b;a; Salimans et al., 2016) or transformers (Parmar et al., 2018; Chen et al., 2020). Kolesnikov & Lampert (2017); Menick & Kalchbrenner (2019) improved performance of autoregressive models with hierarchical modeling (e.g. over color depth or resolution). While obtaining better results than normalizing flows, autoregressive models are also much slower and do not scale to large resolutions as they require a forward-pass per (sub)pixel.

In the context of normalizing flows, autoregressive dependency patterns between latent variables are a popular approach to improve modeling capabilities of normalizing flows (Kingma et al., 2016; Papamakarios et al., 2017; Huang et al., 2018). Bhattacharyya et al. (2020) combined autoregressive modeling with a multiscale architecture. Concurrently to this work, Zhai et al. (2024) proposed a combination of the transformer-based autoregressive flow.

## 5   Conclusion

The Jet model revisits normalizing flows with a focus on simplicity and performance. While eliminating complex components such as multiscale architectures and invertible layers, Jet matches existing normalizing flow models. The architecture with its regular attention mechanism showed significant improvements when paired with transfer learning and establishes new SOTA results, highlighting the need to study these architectures in data-abundant regime which is common for unsupervised methods.

We see normalizing flows, and Jet in particular, as a useful tool for advancing generative modeling. Due to its simple structure and lossless guarantees, it can serve as a building block for powerful generative systems.

One recent example is Tschannen et al. (2024b), which leverages a normalizing flow to enable end-to-end autoregressive modeling of raw high-resolution images. We anticipate more progress in this area and believe that the Jet model will prove itself a powerful normalizing flow component that can be used out-of-the-box for a variety of applications.

**Acknowledgments**

We thank Fabian Mentzer for discussions and feedback on this paper.

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

# A   Appendix

## A.1   Architecture details

Table 3: Architecture details for models used to obtain the main results in Table 1.

|                     | ImageNet-1k | | ImageNet-21k | |
|---------------------|:---:|:---:|:---:|:---:|
|                     | $32 \times 32$ | $64 \times 64$ | $32 \times 32$ | $64 \times 64$ |
| Coupling layers     | 64  | 64  | 64  | 64   |
| ViT depth           | 6   | 5   | 6   | 6    |
| ViT width           | 256 | 512 | 768 | 1024 |
| ViT attention heads | 4   | 8   | 12  | 16   |

## A.2   Comparison with other works

The main text of this work reports numbers in the canonical dataset ImageNet-1k $32 \times 32$ and $64 \times 64$ (Van den Oord et al., 2016b) and also used in the main baseline Flow++ (Ho et al., 2019). There are other works in the literature which deviate from this protocol rendering number comparisons between works hard. Here we compare our work with Denseflow (Grcić et al., 2021) which uses a different downsampled version of ImageNet (Chrabaszcz et al., 2017) and uses data augmentations.

The results in Table 4 show that as know in the community the newer downsampled ImageNet is significantly easier with values significantly lower than in Table 1. Additionally it shows once more that ImageNet-1k is a data constrained setting where data augmentation can play a critical role in better numbers but ultimately it lags significantly behind to using transfer learning.

Table 4: Comparison of Jet with Denseflow

|                          | ImageNet-1k | |
|--------------------------|:---:|:---:|
|                          | $32 \times 32$ | $64 \times 64$ |
| Using augmentation from Grcić et al. (2021) | | |
| Denseflow                | 3.63  | 3.35  |
| Jet                      | 3.638 | 3.355 |
| No data augmentation     | | |
| Jet                      | 3.656 | 3.382 |
| Jet (I21k)               | 3.573 | 3.300 |

## A.3   Detailed results

Table 5: NLL on ImageNet-1k $32 \times 32$ when sweeping architecture hyper-parameters.

| depth | 16 | | | 32 | | | 64 | | |
|-------|:---:|:---:|:---:|:---:|:---:|:---:|:---:|:---:|:---:|
| ViT dim | 256 | 512 | 768 | 256 | 512 | 768 | 256 | 512 | 768 |
| ViT depth | | | | | | | | | |
| 1 | 4.32 | 4.22 | 4.18 | 4.18 | 4.11 | 4.08 | 4.12 | 4.06 | 4.04 |
| 2 | 4.11 | 4.04 | 4.03 | 4.03 | 3.99 | 4.01 | 3.99 | 3.97 | 4.02 |
| 3 | 4.05 | 4.00 | 4.03 | 3.99 | 3.97 | 4.07 | 3.96 | 3.98 | 4.11 |
| 4 | 4.03 | 3.99 | 4.08 | 3.97 | 3.99 | 4.15 | 3.94 | 4.03 | 4.18 |
| 5 | 4.01 | 3.99 | 4.14 | 3.96 | 4.01 | 4.21 | 3.93 | 4.07 | 4.28 |
| 6 | 4.00 | 4.00 | 4.19 | 3.95 | 4.04 | 4.34 | 3.93 | 4.12 | N/A |

Table 6: NLL on ImageNet-1k $64 \times 64$ when sweeping architecture hyper-parameters.

| Couplings | 16 | | | 32 | | | 64 | | |
|---|---|---|---|---|---|---|---|---|---|
| ViT dim | 256 | 512 | 768 | 256 | 512 | 768 | 256 | 512 | 768 |
| ViT depth | | | | | | | | | |
| 1 | 4.02 | 3.94 | 3.90 | 3.92 | 3.84 | 3.81 | 3.85 | 3.79 | 3.77 |
| 2 | 3.86 | 3.78 | 3.75 | 3.78 | 3.72 | 3.70 | 3.74 | 3.69 | 3.68 |
| 3 | 3.81 | 3.73 | 3.71 | 3.74 | 3.69 | 3.68 | 3.70 | 3.66 | 3.69 |
| 4 | 3.79 | 3.71 | 3.69 | 3.72 | 3.67 | 3.68 | 3.69 | 3.66 | 3.70 |
| 5 | 3.77 | 3.70 | 3.69 | 3.71 | 3.66 | 3.69 | 3.67 | 3.66 | 3.73 |
| 6 | 3.76 | 3.69 | 3.69 | 3.70 | 3.66 | 3.70 | 3.67 | 3.66 | 3.75 |

Table 7: NLL on ImageNet-1k $64 \times 64$ when sweeping CNN architecture hyper-parameters.

| Couplings | 16 | | | | | 32 | | | | | 64 | | | |
|---|---|---|---|---|---|---|---|---|---|---|---|---|---|---|
| Block dim | 256 | 512 | 768 | 1024 | 1536 | 256 | 512 | 768 | 1024 | 1536 | 256 | 512 | 768 | 1024 |
| CNN depth | | | | | | | | | | | | | | |
| 1 | 4.17 | 4.05 | 4.00 | 3.96 | 3.92 | 4.05 | 3.95 | 3.91 | 3.87 | 3.83 | 3.97 | 3.88 | 3.84 | N/A |
| 2 | 4.09 | 3.97 | N/A | 3.87 | 3.83 | 3.98 | 3.88 | 3.83 | 3.79 | 3.76 | 3.90 | 3.81 | 3.77 | 3.75 |
| 3 | 4.06 | 3.94 | 3.88 | 3.84 | 3.80 | 3.95 | 3.84 | 3.79 | 3.77 | 3.74 | 3.87 | 3.78 | 3.75 | 3.73 |
| 4 | 4.04 | 3.91 | 3.85 | 3.82 | 3.78 | 3.93 | 3.83 | 3.78 | 3.75 | 3.74 | 3.85 | 3.76 | 5.97 | N/A |
| 5 | 4.02 | 3.90 | 3.84 | 3.81 | 3.77 | 3.91 | 3.81 | 3.77 | 3.75 | 3.74 | 3.84 | 3.76 | 3.98 | N/A |
| 6 | 4.00 | 3.88 | 3.83 | 3.80 | 3.77 | 3.90 | 3.80 | 3.76 | 3.74 | 3.74 | 3.83 | 3.89 | 6.41 | 3.77 |
| 8 | 3.98 | 3.87 | 3.81 | 3.78 | 3.77 | 3.88 | 3.79 | 3.75 | 3.74 | 3.75 | 3.81 | 3.74 | 3.87 | 5.89 |

Table 8: NLL on ImageNet-21k $32 \times 32$ when sweeping architecture hyper-parameters.

| Couplings | 32 | | | 64 | | | 128 | | |
|---|---|---|---|---|---|---|---|---|---|
| ViT dim | 512 | 768 | 1024 | 512 | 768 | 1024 | 512 | 768 | 1024 |
| ViT depth | | | | | | | | | |
| 1 | 3.79 | 3.76 | 3.74 | 3.74 | 3.72 | 3.70 | 3.71 | 3.69 | 3.68 |
| 2 | 3.65 | 3.63 | 3.62 | 3.63 | 3.61 | 3.60 | 3.61 | 3.60 | 3.59 |
| 3 | 3.62 | 3.59 | 3.58 | 3.59 | 3.57 | 3.56 | 3.57 | 3.56 | N/A |
| 4 | 3.59 | 3.57 | 3.57 | 3.57 | 3.55 | N/A | 3.55 | N/A | N/A |
| 5 | 3.58 | 3.56 | 3.56 | 3.55 | 3.54 | N/A | 3.54 | N/A | N/A |
| 6 | 3.57 | 3.55 | 3.54 | 3.55 | 3.53 | N/A | 3.55 | N/A | N/A |

Table 9: NLL on ImageNet-21k $64 \times 64$ when sweeping architecture hyper-parameters.

| Couplings | 32 | | | 64 | | | 128 | | |
|---|---|---|---|---|---|---|---|---|---|
| ViT dim | 512 | 768 | 1024 | 512 | 768 | 1024 | 512 | 768 | 1024 |
| ViT depth | | | | | | | | | |
| 1 | 3.42 | 3.39 | 3.37 | 3.37 | 3.34 | 3.34 | 3.33 | 3.31 | 3.29 |
| 2 | 3.29 | 3.26 | 3.24 | 3.26 | 3.23 | 3.24 | 3.23 | 3.20 | 3.19 |
| 3 | 3.25 | 3.22 | 3.20 | 3.22 | 3.19 | 3.21 | 3.19 | 3.16 | N/A |
| 4 | 3.23 | 3.19 | 3.17 | 3.19 | 3.16 | 3.15 | 3.16 | 3.14 | N/A |
| 5 | 3.21 | 3.18 | 3.16 | 3.18 | 3.15 | N/A | 3.15 | N/A | N/A |
| 6 | 3.20 | 3.17 | 3.15 | 3.17 | 3.14 | 3.13 | 3.14 | N/A | N/A |

Table 10: NLL on ImageNet-1k $64 \times 64$ when varying the ratio between channel and spatial couplings.

| Couplings ratio | Type of spatial couplings | | | |
|---|---|---|---|---|
| (channel : spatial) | row-wise | column-wise | checkerboard | mix of all |
| all spatial | 3.877 | 3.882 | 3.872 | 3.787 |
| 1:1 | 3.747 | 3.742 | 3.741 | 3.737 |
| 2:1 | 3.735 | 3.731 | 3.731 | 3.725 |
| 3:1 | 3.732 | 3.729 | 3.728 | 3.723 |
| 4:1 | 3.731 | 3.726 | 3.724 | 3.719 |
| 5:1 | 3.728 | 3.726 | 3.723 | 3.720 |
| all channel | 3.742 | | | |

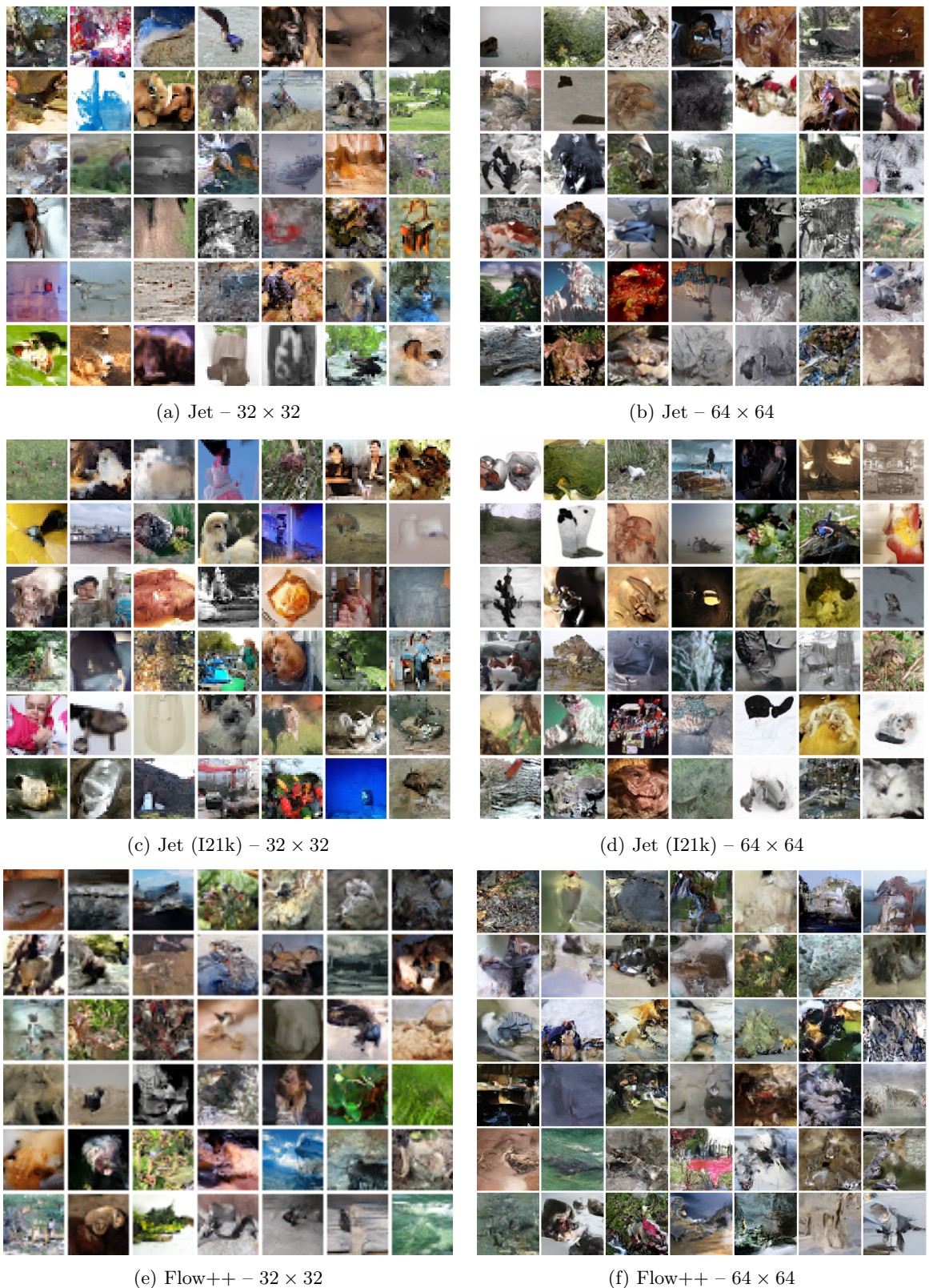

(a) Jet − 32 × 32

(b) Jet − 64 × 64

(c) Jet (I21k) − 32 × 32

(d) Jet (I21k) − 64 × 64

(e) Flow++ − 32 × 32

(f) Flow++ − 64 × 64

Figure 5: Random samples for ImageNet-1k at both 32 × 32 and 64 × 64 resolution. We show samples from Jet when trained from scratch and when finetuning a model pretrained on ImageNet-21k. For comparison we also show samples from Flow++ (Ho et al., 2019).

