# OpenReview forum: "Jet: A Modern Transformer-Based Normalizing Flow"
_TMLR — Accepted by TMLR_

### Review · Reviewer_9Uwh · 2025-01-23

**Summary Of Contributions:**

This paper improves upon previous (one step) normalizing flow models by imbuing them with more recent architectural changes.

**Audience:**

Yes

**Claims And Evidence:**

Yes

**Requested Changes:**

See weaknesses.

**Strengths And Weaknesses:**

Strengths
-----------
This paper goes into great detail about the architectural + other ablations, which serves as a powerful artifact for anyone interested in applying recent architectures (e.g. vision transformers) to flow models.

Weaknesses
---------------
The major weakness is that the comparison on imagenet32x2 and 64x64 is a bit suspect. The paper claims to compare on the right dataset of Crabaszcz et al, but the canonical reference is Van der oord et al 2016 (pixel-rnn/cnn paper). This can be fine if the baselines also use the same dataset, but it seems that the flow++ paper uses Van der oord, so this needs to be clarified.
There is also a question about the baselines used. In particular, Flow++ is 6 years old, which warrants a deeper look into more recent architectures. For example, the paper "densely connected normalizing flows" should satisfy the baseline requirements as a one step architecture and should be compared with (but note that this reference does not use the right/comparable dataset either!).

---

> ### Author Response · Authors · 2025-01-24
> **Clarifications**
>
> > The paper claims to compare on the right dataset of Crabaszcz et al, but the canonical reference is Van der oord et al 2016 (pixel-rnn/cnn paper). This can be fine if the baselines also use the same dataset, but it seems that the flow++ paper uses Van der oord, so this needs to be clarified.
>
> Thank you for pointing this out. It is a typo that we have introduced when looking for a canonical reference for downsampled imagenet. We, in fact, use the downsampled imagenet data through TensorFlow Data Sets https://www.tensorflow.org/datasets/catalog/downsampled_imagenet, which uses the data from http://image-net.org/small/download.php as indicated in the canonical "pixelcnn/pixelrnn" reference. Note the the latter link is not live at the moment. We have, however, pre-downloaded the data when the link was still functional.
>
>
> > There is also a question about the baselines used. In particular, Flow++ is 6 years old, which warrants a deeper look into more recent architectures. For example, the paper "densely connected normalizing flows" should satisfy the baseline requirements as a one step architecture and should be compared with (but note that this reference does not use the right/comparable dataset either!).
>
> We compare to the best normalizing flow baseline available in the literature. "densely connected normalizing flows" paper reports non-comparable evaluation results on Imagenet 32x32 and 64x64, because it does not use the correct downsampling algorithm. This is discussed (and admitted) by the paper author in the official paper's github repo: https://github.com/matejgrcic/DenseFlow/issues/7. The paper also departs from the standard protocol of not doing training-time data augmentations (it does horizontal flips, see Appendix C), which further leads to inflated NLL estimate and even more unfair comparison.
>
> We appreciate your in-depth look into eval methodology and hope our reply addresses your concerns.

---

> ### Comment · Reviewer_9Uwh · 2025-02-02
> **Thanks for Clarification**
>
> Thanks for clarifying the point about the dataset. That is the right one and is a fair comparison.
>
> If possible, I would appreciate at least some more recent method (one can run the preexisting code and compare directly). Although not strictly necessary, this would help with benchmarking. I understand reservations with something like densely connected normalizing flows, but the premise ultimately holds more generally.

---

> > ### Author Response · Authors · 2025-02-22
> > **Additional comparison to the DenseFlow model**
> >
> > > I would appreciate at least some more recent method (one can run the preexisting code and compare directly)
> >
> > We have additionally compared directly to the DenseFlow model. To achieve direct comparison, we retrain Jet on the downsampled ImageNet (Chrabaszcz et al., 2017) version used in the DenseFlow paper, and also follow their data augmentation strategy. Otherwise, we reuse exactly the same hyper-parameters as used in the Jet model paper, without any retuning, and observe the following results:
> >
> > | Dataset     | Our Result   | DenseFlow Paper   |
> > |:------------|:-------------|:------------------|
> > | ImageNet 32 | 3.638 bpd    | 3.63 bpd          |
> > | ImageNet 64 | 3.355 bpd    | 3.35 bpd          |
> >
> > From this we conclude that Jet matches DenseFlow, while being much simpler: no ActNorm layer, no invertible Dense layer, no multiscale architecture, no dense connectivity structure, and no specialized Nyström Self-Attention. We also observe again that the ImageNet setup is data constrained and that one can improve the results with data augmentation and even more with transfer learning. We will add an appendix section about this comparison.

---

### Review · Reviewer_HWBQ · 2025-02-05

**Summary Of Contributions:**

This paper investigates the potential benefits of using new transformer based architectures (ViT) for normalizing flows, a recently neglected class of generative models. The proposed method significantly simplifies the model architecture without the numerous engineered components found in previous normalizing flow methods, only relying on the patch-based attention of the ViT architecture. Evaluations are performed on standard image generation tasks, using a standard metric.

**Audience:**

Yes

**Claims And Evidence:**

Yes

**Requested Changes:**

Generally, the style of the paper appears to be of a technical report that covers the results of an extensive hyper-parameter sweep.
Although the proposed approach shows significant improvement across the range of tested hyper-parameters, the scope is very limited.
There is limited discussion on limitations of the method, the scope makes it difficult to even imagine how the method may be applied beyond a toy setting, and the presentation/discussion of the results does not convey how the work fits into the greater space of work on generative models.

I recommend adding final evaluations using a metric that is not NLL (FID) to further strengthen claims about model performance. At minimal a discussion, if not new experiments, to investigate limitations and scaling properties of the method would be highly valuable. The inclusion of final quantitative results for other relevant methods along with some discussion would help place the work in the space of other generative modelling work.

The transfer learning experiments contributes little to the primary claims of the paper, potentially moving these into the appendix would improve the coherence of the paper. Unless there is a way to make these experiments more relevant that I have not thought of.

**Strengths And Weaknesses:**

__Strengths__

Although normalizing flows have recently fallen out of favour for many generative tasks, they have the benefit of producing exact likelihoods, while being fast to sample from. Few work exists that evaluates the use of this specific architectural advancement (ViT) for this class of models. The proposed method is far simpler in terms of model complexity, showing these ViT base architectures can outperform previous methods with a simpler design space. Standard datasets (Imagenet, CIFAR-10) are used for evaluation, while performance is measured using negative log likelihood (NLL), also a standard metric. Comparisons with a state-of-the-art flow model shows a favourable performance for the proposed method. Extensive ablations support the claim of improved performance over a wide range of hyper-parameters.


__Weaknesses__

Although the claims are mostly supported, the scope of the method is potentially very general while the scope of the experiments is narrow.

The only metric considered is NLL. While this is a standard metric in this space, NLL is known to not always correlate with image quality for image generation tasks. More recent work on normalizing flows often include FID to further support claims about model performance.

The major contribution of the paper is the empirical validation of a ViT architecture for normalizing flows.
The experiments are limited to toy vision datasets. Although these are standard, they leave questions about the performance of the method beyond a toy setting. This can be important since transformers have quadratic complexity, but the images considered are relatively small and may not reveal scaling properties of the model. Other related factors that may be of interest is the effect of patch size, number of tokens, and different data modalities.

Only one baseline primarily used for comparison since it is the state-of-the-art. However, it would be instructive to include qualitative comparisons with other generative models to provide the reader a greater context of how the proposed method fits into the space of generative modelling. Often, normalizing flow methods are compared with autoregressive likelihood methods, VAEs, and even diffusion models, since these methods can also provide exact likelihoods or at least approximations for NLL, while also potentially sharing architectural similarities. Although this is the only work to my knowledge that specifically investigates a ViT architecture for normalizing flows, it is not the only to investigate the use of transformers.

There are also few limitations discussed, making the paper seem one sided, making it difficult for the reader to determine applicability of the model outside of toy settings, or potential directions of future research.

Transfer learning is also investigated to mitigate overfitting, but seems to be an orthogonal to the main contributions of the paper. This approach to prevent overfitting is a general method not specific to normalizing flows, and does not seem to add to the contributions.

---

> ### Author Response · Authors · 2025-02-22
>
> > I recommend adding final evaluations using a metric that is not NLL (FID)
>
> Following the protocol of the DenseFlow paper, we conducted FID evaluation of the Jet model on the CIFAR-10 dataset. Jet achieves 27.4 FID, while DenseFlow obtains 34.9 FID. We expect that, despite similar NLL between Jet and DenseFlow, a better FID metric for the Jet model stems from the fact that Jet uses a global attention mechanism which helps to better capture high-level image dependencies.
>
> > The transfer learning experiments contributes little to the primary claims of the paper
>
> We believe that transfer learning results constitute an important contribution to the paper.
>
> Jet model is generally more flexible and expressive than its predecessors: it has fewer inductive biases (e.g. no multiscale components), but instead relies on the very simple regular architecture powered by the attention mechanism. To be able to fully leverage Jet’s potential, we explore the transfer learning angle.
>
> It is not obvious that a Flow model pretrained on different data will transfer well to the target dataset. For example, there is a large domain gap between CIFAR-10 images and ImageNet-21k images. Importantly, we demonstrate the effectiveness of transfer learning in this setting.

---

### Review · Reviewer_bjSh · 2025-02-08

**Summary Of Contributions:**

The paper introduces a model called Jet. It is a normalizing flow model with coupling layers parametrized by vision transformer. The authors aim to simplify the model architecture while improving the performance. Jet does not use many advanced techniques developed for normalizing flows, while demonstrating competitive performance.

**Audience:**

Yes

**Claims And Evidence:**

No

**Requested Changes:**

* Tone down state-of-the-art claims.
* Report number of parameters, memory requirements and sampling time compared to previous works.
* What is the reasoning behind plain CNN architecture as a baseline and not a model with Self-Attention (as done in Flow++)? According to Figure 1 in Flow++ paper, Attention improves over the standard CNN. I think this would be a better baseline to have.


* Writing and missing details:
   * For some reason non-finetuned result for CIFAR10 are missing. Please, add them.
   * Result for ablation study in section 3.2.6 are not reported, please fix it.
   * Move Figure 2 to experimental section.
   * I think Figure 3 is missing the colour-coding, similarly to Figure 2. It is hard to make any useful conclusions from it currently.
   * Please, clarify how the dimension splitting is implemented. According to 2.1 patches are flattened so that the input has shape $K \times 2d$. Is the last dimension in section 2.1 the same as "spacial dimension" mentioned in section 2.5? In this case, is spacial splitting performed before the flattening of the patches or after? Does channel-wise splitting split each patch channel-wise or is it split over the "patch" dimension?
   * Section 3.2.4. Paragraph 1 talks about "K patches", paragraph 2 talks about "N/2 patches". I assume K and N are the same thing. Please, standardize the terminology to avoid confusion.

**Strengths And Weaknesses:**

**Strengths**
* Authors conduct a wide range of ablation studies
* The paper demonstrate the advantage of pre-training

**Weaknesses**
* The performance improvements are modest unless pre-training is employed. Jet outperforms Flow++ only with pre-training. For a fair comparison of the pre-trained model, one would need to pre-train Flow++ on Imagenet-21K as well. There also some better results reported on these datasets with normalizing flows (e.g. ANF [1] and VFlow [2]). At the same time, authors repeatedly mention "state-of-the-art" performance.

[1] Huang, Chin-Wei, Laurent Dinh, and Aaron Courville. "Augmented normalizing flows: Bridging the gap between generative flows and latent variable models." arXiv preprint arXiv:2002.07101 (2020).

[2] Chen, Jianfei, et al. "Vflow: More expressive generative flows with variational data augmentation." International Conference on Machine Learning. PMLR, 2020.

* Model size and memory consumption are not reported, but form the hyperparameters values I assume it is much more expensive to train and sample from Jet compared to Flow++.

* The paper is not clearly written in several place. Please, see the requested change for the details.

---

> ### Author Response · Authors · 2025-02-22
>
> > The performance improvements are modest unless pre-training is employed
>
> This is true: Jet models have much fewer inductive biases than prior architectures and generally require more data to achieve state-of-the-art performance.
>
> In the paper we demonstrate that transfer learning between very different image datasets (e.g. ImageNet-21k and CIFAR-10) is very effective. At the same time, unsupervised image data is abundant. This means that in practice there is no reason not to use more general models with pretraining.
>
> In general, this is very reminiscent of CNN vs ViT models for perception tasks. While ViTs generally require more data, they dominate CNNs when pretraining on web-scale data is employed, see [“Learning Transferable Visual Models From Natural Language Supervision” by Radford et al.](https://arxiv.org/abs/2103.00020) as an example.
>
> > Comparison to ANF [1] and VFlow [2]
>
> Both of these references represent a different model class and can be seen as an orthogonal line of research about incorporating latent variables in flow-based models. Jet is compatible with these methodologies but integrating latent variables in Jet falls out of scope of this paper. Furthermore, both of these works rely on maximizing an ELBO, rather than directly minimizing NLL as done for standard feed-forward flows such as Jet, which is a crucial difference.
>
> > it is much more expensive to train and sample from Jet compared to Flow++
>
> This is not the case.
>
> Flow++ report inference speed for their model: 320ms for a batch of 8 images with resolution 32x32. We run an equally capable Jet model (32 coupling layers, ViT depth 3, embedding size 512 that matches Flow++ on the downsized imagenet64 benchmark) for an equally-sized batch of images, and measure 23ms inference speed, which is more than 10x faster.
>
> One caveat is that Flow++ uses NVIDIA 1080 Ti GPU, and we do not have access to such hardware. We did our best effort to match 1080 Ti computational power and used TPU-v2 for our benchmarking. TPU-v2 has peak performance of 22.5 TFlops, while 1080 Ti GPU has peak performance of 11.3 TFlops. So, overall, we expect Jet to be 5 times faster than a similarly capable Flow++ model.
>
> Note, we intentionally do not compare parameter counts as it is known to be a very misleading metric, see [“The Efficiency Misnomer” by Dehghani et al.](https://arxiv.org/abs/2110.12894).
>
> > tone down state-of-the-art claims
>
> We agree with the reviewer that SOTA claims need to be more precise. We will make it clear in the paper that SOTA is achieved in the transfer learning setting.
>
> > What is the reasoning behind plain CNN architecture as a baseline and not a model with Self-Attention (as done in Flow++)? According to Figure 1 in Flow++ paper, Attention improves over the standard CNN. I think this would be a better baseline to have.
>
> We observe that a simple regular ViT-based flow already matches and outperforms competing models when sufficient data is provided.
>
> In the data-limited regime (like CIFAR-10 dataset), we expect hybrid architecture to be the optimal. However, there is little research motivation to invest resources into this question: unsupervised training data is abundant and transfer learning works really well, as we show. For this reason a thorough exploration of hybrid architectures did not enter the scope of the project.
>
> > Please, clarify how the dimension splitting is implemented. According to 2.1 patches are flattened so that the input has shape K x 2d
>
> When we say “channel” splitting we refer to splitting each patch channels into two groups of channels, which transforms `K x 2d` array to two `K x d` arrays
>
> When we say “spatial” splitting we refer to splitting patches, which will result in two `K/2 x 2d` arrays. Patch split is performed after flattening, however, it respects the original 2D structure of the patches (e.g. “checkerboard” split).
>
> > I think Figure 3 is missing the colour-coding
>
> Figure 3 shows scaling trends: as compute is increased, the flow models improve assuming sufficient data to mitigate overfitting is available. To highlight that we will add the train NLL in the figure. Color coding is indeed useful to highlight most salient additional observations, which we do in a separate plot (Figure 2), to avoid an overwhelming information density in Figure 3.
>
> > Writing suggestions
>
> Thank you for the suggestions, we will fix them in the next revision.

---

### Decision · Action_Editor_657D · 2025-03-17

**Recommendation:** Accept with minor revision

**Comment:**

This paper proposes a new Transformer-based architecture, Jet, for normalizing flows. The reviewers are in consensus on being a bit on the fence with regards to the significance of the findings of the paper. After having read through the paper, I am of the opinion that the acceptance can be defended:

* the topic is of general interest
* the paper makes a thorough scaling analysis of the investigated architectures (Jet and CNN baseline)
* the paper gives and in depth description and analysis of the model architecture.

The reviewers have some additional suggestions for improvements, some of which the authors have already agreed to do. I encourage the authors to make these because this will improve the usefulness and impact of the paper.

**Audience:**

Yes.

**Claims And Evidence:**

The reviewers have some concerns about claims and evidence:
- SOTA - The authors response "We agree with the reviewer that SOTA claims need to be more precise. We will make it clear in the paper that SOTA is achieved in the transfer learning setting."
- Two reviewers are not convinced that studying transfer learning is enough "I believe it is necessary for this paper to explore performance gains at scale to be of interest to readers, since this is the primary benefit of these ViT type models." The authors are encouraged to add this to the paper.